# Effects of Preceding Transcranial Direct Current Stimulation on Movement Velocity and EMG Signal during the Back Squat Exercise

**DOI:** 10.3390/jcm11175220

**Published:** 2022-09-03

**Authors:** Manuel Garcia-Sillero, Iván Chulvi-Medrano, Sergio Maroto-Izquierdo, Diego A. Bonilla, Salvador Vargas-Molina, Javier Benítez-Porres

**Affiliations:** 1Faculty of Sport Sciences, EADE-University of Wales Trinity Saint David, 29018 Málaga, Spain; 2Sport Performance and Physical Fitness Research Group (UIRFIDE), Department of Physical and Sports Education, University of Valencia, 46010 Valencia, Spain; 3Department of Health Sciences, European University Miguel de Cervantes (UEMC), 47012 Valladolid, Spain; 4Research Division, Dynamical Business & Science Society—DBSS International SAS, Bogotá 110311, Colombia; 5Research Group in Physical Activity, Sports and Health Sciences (GICAFS), Universidad de Córdoba, Montería 230002, Colombia; 6Sport Genomics Research Group, Department of Genetics, Physical Anthropology and Animal Physiology, Faculty of Science and Technology, University of the Basque Country (UPV/EHU), 48940 Leioa, Spain; 7Physical Education and Sport, Faculty of Medicine, University of Málaga, 29016 Málaga, Spain

**Keywords:** neuromuscular manifestations, neurofeedback, surface electromyography, linear position sensor, strength training

## Abstract

This study aimed to evaluate the effects of preceding anodal transcranial direct stimulation (a-tDCS) over the dorsolateral prefrontal cortex (DLPFC) during the back squat exercise on movement velocity and surface electromyographic (sEMG) activity. Thirteen healthy, well-trained, male firefighters (34.72 ± 3.33 years; 178 ± 7.61 cm; 76.85 ± 11.21 kg; 26.8 ± 4.2 kg·m^−2^; back squat 1-repetition maximum 141.5 ± 16.3 kg) completed this randomised double-blinded sham-controlled crossover study. After familiarisation and basal measurements, participants attended the laboratory on two occasions separated by 72 h to receive either Sham or a-tDCS (current intensity of 2 mA for 20 min). Immediately after stimulation, participants completed three sets of 12 repetitions (70% of 1-RM) with three minutes of recovery between sets monitored with a linear position transducer. The sEMG of the rectus femoris (RF) and vastus lateralis (VL) of both legs were recorded. No significant differences were observed between a-tDCS and Sham interventions on mean concentric velocity at any set (*p* > 0.05). Velocity loss and effort index were significantly higher (*p* < 0.05) in set 3 compared to set 1 only in the a-tDCS group. The right-leg RM and right-leg VL elicited the greatest muscle activation during set 1 after a-tDCS and Sham, respectively (*p* < 0.05). Our results revealed that a-tDCS over the DLPFC might impact movement velocity or fatigue tolerance in well-trained individuals. Notwithstanding, significant differences in dominant-leg muscle activity were found both in a-tDCS and Sham.

## 1. Introduction

To optimise strength training, new promising techniques such as transcranial direct current stimulation (tDCS) are available [1]. This tool has been used in the field of neuro-rehabilitation to improve motor skills and increase the excitability of the supplementary motor area [2]. However, recent research has focused on the use of tDCS as a coadjutant/complementary tool to boost neuromuscular adaptations after strength or endurance training [3]. tDCS is a non-invasive neuromodulation brain stimulation technique that continuously emits a low current through the target brain area [4]. tDCS can change the transmembrane potential of neurons, thus affecting the level of excitation, and regulating the firing rate of isolated neuronal cells [5]. Over a period of 3 to 20 min, a direct current of 0.4 to 2.0 mA is applied to the relevant area of the cortex [6]. Although this electrical stimulus can be applied in different areas of the cerebral cortex, the most studied for improving muscle strength are the primary motor cortex (M1) and the dorsolateral prefrontal cortex (DLPFC) [7]. During this process, significant changes in cortical excitability have been demonstrated to be achieved [8]. Different effects might be attained depending on the polarity of the selected current, which include increased excitability (using anode; a-tDCS) or decreased excitability (using cathode; c-tDCS) of the cortex in the target area [9].

On the other hand, the control of movement velocity has lately been proposed as a valid alternative way to monitor training load when seeking training optimisation [10,11]. The influence of movement velocity modulation during strength exercise on neuromuscular fatigue [12] and on the adaptive capacity of the neuromuscular system has been widely investigated [13,14]. In this regard, recent research has demonstrated the importance of movement velocity control as a fundamental metric for weight resistance training (RT) [14] and fatigue assessment [15]. It is well-known that the progressive neuromuscular fatigue, denoted by velocity loss, is an important variable in the configuration of the strength training program, as it leads to a wide range of different neuromuscular adaptations [16].

Complementary, surface electromyography (sEMG) devices have also been used to monitor fatigue during exercise [17,18]. Indeed, a-tDCS has shown positive effects in this regard [19]. Other effects of tDCS on physical-performance-related variables, such as jumping and sprinting ability, have been demonstrated recently [20]. Improvements have also been found in endurance tests after the application of tDCS; for instance, Park et al. [21] showed an increase on the running time to exhaustion in participants who were given a stimulation for 20 min at 1.98 mA. Interestingly, a-tDCS applied to the motor cortex before exercise has been shown to produce acute and chronic increases in muscle strength not only during bilateral exercises, but also unilateral exercises [22]; the use of maximal voluntary isometric contraction as an indicator of neuromuscular performance has not shown positive effects [23]. In addition, it has been observed that the pre-training application of a-tDCS over the dorsolateral prefrontal cortex might increase training volume during an upper-limp strength exercise program [19]; albeit no significant changes were observed in the 1-repetition maximum (RM) or in the force–velocity relationship parameters, the a-tDCS allowed the maintaining of higher movement velocities while reducing the rate of perceived exertion (RPE) [19]. In this sense, the correlations between the effort index (EI) and the RPE as fatigue indicators provide a better understanding of the real degree of effort made during strength exercise [24]. The current scientific literature still shows contradictory results regarding the effect of this technique on skeletal muscle performance; thus, this study aimed to evaluate and clarify the effects of pre-training a-tDCS on muscle-strength-related variables (i.e., movement velocity, velocity loss, and sEMG signal) in highly trained men.

We hypothesised that the application of a-tDCS before training would result in the increase of exercise movement velocity and muscle activity in both legs during several sets of back squats (BS) at 70% of 1-RM.

## 2. Materials and Methods

### 2.1. Participants

A total of 16 healthy male firefighters from the Royal Official Fire Brigade of Malaga were potentially eligible to participate in this research. None of the participants had previous lower-limb injuries during the six months prior to the study, had more than two years of experience in strength training, and were right-leg dominant (self-reported appearance). Besides the Physical Activity Readiness Questionnaire (PARQ), all participants were assessed by means of a physical examination and a physical activity habits questionnaire, the IPAQ [25]. All participants showed high physical activity values and minimal risk for physical activity through these questionnaires. Three participants did not complete the measurements due to personal situations; therefore, thirteen completed the study and were randomised, analysed, and reported in this article. Table 1 shows the characteristics of the participants.

Participants were instructed to avoid taking analgesics (6 h), alcohol (48 h), and caffeine (6 h), and to avoid strenuous exercise (48 h) before each session. In addition, they were informed about the general experimental procedures and risk factors related to the measurement, and the research objectives and hypotheses. All tests were performed at Eshmún Sport Clinic (Málaga, Spain). The research protocol was reviewed and approved by the Ethics Committee of the EADE-University of Wales Trinity Saint David (Málaga, Spain) Committee’s reference number: EADECAFYD2020-2, in accordance with the ethical guidelines of the Helsinki Declaration [26].

### 2.2. Trial Design

A randomized, double-blinded, sham-controlled crossover study was performed to evaluate the effects of a-tDCS on muscle-strength-related variables. Sham stimulation is recommended to evaluate the efficacy of the active stimulation and placebo effects [27]. All participants visited the laboratory in three separated times. During the first visit to the laboratory, the familiarisation session consisted of measurement of the anthropometry-related variables (body mass and stature) of the participants, followed by the BS exercise (1-RM) test. During the second and third session, the a-tDCS or the Sham protocol were randomly applied (i.e., crossover design). There was a time period of five days between both intervention protocols (Figure 1).

#### 2.2.1. Back Squat One-Repetition Maximum Test (BS 1-RM)

The warm-up consisted of five minutes of stationary cycling (Artis Bike Standar, Technogym Trade Company 2017, Cesena, Italy) at a gentle, self-selected pace. In addition, five minutes of lower-limb joint mobilisation exercises and three 30-m running accelerations were performed. Subsequently, the participants carried out 3 sets of 5 repetitions of BS with fixed weights of 20, 30, and 40 kg. The 1-RM test was performed in a multipower device (Element + Multipower, Technogym Trade Company 2017, Cesena, Italy) using a progressive load test [28]. The initial load was set at 20 kg, and was progressively increased until the average propulsive velocity achieved was less than 0.70 m∙s^−1^ [28]. Thereafter, the load was adjusted individually using small increments (from 2.5 to 5 kg), and the 1-RM was estimated when participants lifted the heaviest load while completing the full range of motion (90°) and without any external assistance. The depth of the squat was determined for each participant using a Halo digital goniometer (Halo Medical Devices, Subiaco, Western Australia) to ensure that the knee angle was 90°. The range of movement (ROM) was ensured during the tests by placing a bench adapted to each subject as a reference. Three attempts were performed for the light loads (speed > 1.15 m∙s^−1^), 2 for medium (0.70–1.15 m∙s^−1^), and only one for heavier loads (<0.70 m∙s^−1^), always focusing on the concentric phase of the exercise. The rests between sets were 3 min for light and medium loads, and 5 min for heavier loads. The best repetition at each load was considered according to the criterion of the mean propulsive velocity [29]. In addition, EI, as an indicator of fatigue, was measured for each condition (a-TDCS and Sham) according to previous studies [24].

#### 2.2.2. Back Squat Exercise Protocol (BS)

The same multipower device (Element + Multipower, Technogym Trade Company 2017, Cesena, Italy) used during the 1-RM test was used to perform the BS exercise. Three sets of 12 repetitions were performed by each participant with a 3-min rest period between sets [12]. The exercise was started from an upright position, with knees and hips fully extended, approximately shoulder-width apart, and with both feet placed on the floor in parallel to each other. Strong verbal encouragement and speed feedback was provided to motivate participants to give maximum effort and the highest possible performance in the concentric phase of the movement. They were not allowed to take their feet off the ground, although they were allowed to lift their heels at the end of the concentric phase. The same warm-up protocol used before the 1-RM test was applied before each experimental session. The mean concentric velocity of each set (average of 12 repetitions), the mean propulsive velocity of each set (average of 12 repetitions), the peak velocity of each set, the concentric velocity loss from the fastest (usually first) to the slowest (last) repetition of the set, and the effort index (calculated by multiplying the concentric velocity loss of the set by the mean concentric velocity of the best repetition) were the kinematic variables collected for further analysis. The participants spent 5 min from the time they finished receiving the a-tDCS stimulus and started this protocol.

#### 2.2.3. a-tDCS Procedures

Before exercise intervention, each participant was placed in the supine position for 10 min on the laboratory couch, a position they maintained during the a-tDCS stimulation. A pair of pads immersed in a 140 mM NaCl solution containing two electrodes (7 × 5 cm, 35 cm^2^) were used to apply a 2 mA current for a total stimulation time of 20 min (ramp-up/down time of 30 s each), as has been described before [8]. The electrodes (anode and cathode) were connected to a direct current stimulation device (EPTE Bipolar System, Valencia, Spain). For a-tDCS, according to the electroencephalogram of the international 10–20 EEG system [30], the anode electrode was placed on the left DLPFC located in the F3 electrode area, whereas the cathode was placed on the frontal cortex above the right eye (in the area of electrode Fp2) and fixed in place using elastic bands. For sham tDCS, the participants received an initial electrical stimulation (ramp-up/down time of 15 s each), but no other stimulus was received. According to previous studies, this procedure allows the participant to perceive a tingling or prickling sensation that mimics the initial stimulus of the a-tDCS [31]. In this sense, participants were unaware of the type of stimulus they would receive during the test, and a placebo effect could be achieved [32].

#### 2.2.4. Surface EMG Protocol

sEMG data, as root mean square (EMGrms), were collected from the quadriceps (rectus femoris (RF) and vastus lateralis (VL)). The SENIAM project recommendations were followed for skin preparation and electrode placement on RF and VL [33]. The skin was prepared by shaving the hair with a razor blade, and then cleaning with cotton and 70% alcohol, rubbing until a reddish colour was achieved to improve electrode adherence, and allowing it to volatilise before being placed. The electrodes were placed with the subject in a seated position and the knee slightly flexed (distance between electrodes: 20 mm). The same researcher was always in charge of mounting the electrodes to ensure the same location of the electrodes. A wireless sEMG telemetry system (mDurance Solutions SL, Granada, Spain) was used to collect the data. The mDuranceR© system is a bipolar EMG system composed of 4 EMG channels with a sampling frequency of 1024 Hz and a bandwidth of 8.4 kHz. The resolution of the EMG signal is 24 bits, and the overall amplification is 100–10,000 *v*/*v*. The electrodes used were pre-gelled Ag/AgCl with a diameter of 10 mm. A reference electrode was placed on the head of the fibula of the same leg [34].

Signals were filtered using a fourth-order Butterworth bandpass filter with a cut-off frequency at 20–450 Hz. The signal was smoothed using a window size of 0.025-s RMS, and an overlapping of 0.0125 s between windows [35]. Before the start of the squat tests in each of the sessions, the basal level of the sEMG signal was monitored to check that there were no differences in this value. During the sessions, the electrodes were placed on the subject at the same time as the a-tDCS assembly, before the exercise. The recording of sEMG data was synchronised with the repetitions and the times for the start and end position with the position transducer, recording each and every repetition performed by each participant.

### 2.3. Statistical Analyses

Statistical analyses were performed using SPSS v.26.0 (IBM Corp., Armonk, NY, USA). The results were expressed as mean ± SD. Data distribution was examined for normality using the Shapiro–Wilk test. A repeated-measures analysis of variance (ANOVA) with one between-subjects factor (intervention group, i.e., a-tDCS or Sham) and one within-subjects factor (set (set 1, set 2, and set 3)) followed by Bonferroni post-hoc tests to investigate differences in the variables measured were performed. Additionally, to investigate possible differences between legs in muscle activity, a 3-way ANOVA, including a second within-subjects factor leg (right and left) was performed. The significance level was set to *p* < 0.05. The sample size was estimated using data from previous studies in which a-tDCS was investigated for its effect on kinematic variables during resistance exercise. Based on the effect size of 0.3 for a possible difference in mean concentric velocity during exercise between the a-tDCS and Sham conditions, it was estimated that at least 10 participants were necessary, with an alpha level of 0.05 and a power (1−β) of 0.80 by G*Power (G*Power 3.1.9.2, Heinrich-Heine-Universitat Dusseldorf, Dusseldorf, Germany; http://www.gpower.hhu.de/, accessed on 3 November 2020).

## 3. Results

The statistical analysis showed no significant effects (*p* > 0.05) for the intervention group, set, or intervention × set interaction on mean concentric velocity (MV). Hence, no significant differences were observed between a-tDCS and Sham interventions on MV in any set (Figure 2). Similarly, when mean propulsive velocity and mean concentric peak velocity were analysed, no significant interactions were found either between interventions or between sets. Regarding concentric velocity loss (Figure 3) and EI (Figure 4), a significant main effect of set was found, showing significant differences between set 1 and 3 (*p* < 0.01, F = 5.95; and *p* < 0.05, F = 4.91, respectively). Thus, a higher concentric velocity loss (*p* < 0.05, 35.2%) and a higher EI (*p* < 0.05, 33.0%) were observed in the a-tDCS condition when set 3 was compared to set 1. However, no significant differences were observed between a-tDCS and Sham conditions.

The muscle activity (EMGrms) in RF and VL during the BS exercise for each set and leg are shown in Table 2. Statistical analysis revealed a significant main effect of set for both RF and VL EMGrms (F range 6.2–8.7). In addition, there was a condition x*set interaction in VL, showing differences between set 1 and set 2 in the a-TDCS condition (*p* < 0.05, F = 5.41), and between set 2 and 3 in the Sham condition (*p* < 0.05, F = 3.95). However, there was no condition x*set, condition x*leg, or set x*leg interaction in any of the RF EMGrms assessments. However, statistical analysis revealed a significant main effect of set for both RF and VL EMGrms (F range 6.2–8.7). Thus, significant differences between a-tDCS and Sham conditions were observed in set 1 and set 2, in which right RF showed higher EMGrms values (*p* < 0.01, 51.1%; and *p* < 0.01, 30.8%, respectively). No significant differences between legs nor between sets were observed for any variable.

## 4. Discussion

This study was designed to further explore the potential ergogenic effect of a-tDCS preceding RT exercises. Considering the absence of information on the effects of a-tDCS on the movement velocity, this study aimed to evaluate the acute effects of pre-training a-tDCS over the DLPFC on exercise movement velocity and lower-limb muscle activation (rectus femoris and vastus medialis) during the BS exercise. The main finding of the study was the no significant differences on movement velocity between a-tDCS or Sham conditions. No significant effects for any intervention (i.e., a-tDCS or sham protocols), set, or intervention x set interaction on mean concentric velocity were found. Although no significant differences were observed between a-tDCS and Sham interventions at any set, the a-tDCS group experienced higher concentric velocity loss and higher EI values when comparing set 3 to set 1, an aspect that can be of great application in certain areas such as injury rehabilitation. The depolarising neural soma and hyperpolarised apical dendrite may explain this [36], but we are aware that more research is warranted.

Overall, the previous publications on a-tDCS have
provided data pointing to the ergogenic effects on exercise performance in
healthy individuals. Even though the application time (15–20 min), the
intensity (≈2 mA), and the predominant application areas (M1 and DLPFC) are
very standardized, there is a great heterogeneity in the applied exercise
methodology. This inconsistency in the results could also be attributed to the
fact that the main physiological mechanisms involved are unknown. The
facilitation of the primary motor cortex by increasing its output during
exercise and possibly reducing supraspinal fatigue has been suggested [3]. The findings of this study are contrary to
Lattari et al. [7], who reported an increase
in the number of repetitions during a single set of the leg press exercise
performed to muscle failure immediately after the application of a-tDCS,
suggesting that it could be an effective tool to increase training volume in
strength exercises [19]. Previous
investigations have also shown positive effects in endurance tasks [37], especially in submaximal tasks of a cyclic
nature, such as time until failure when running at 80% of maximum intensity [21]. In fact, a recent study [19] showed improvements in the total number of
repetitions performed at 75% of 1-RM with one minute of recovery, besides
improvement in the velocity loss and strain rate. This suggests the possibility
of delaying the onset of fatigue and loss of performance in strength-related
tasks after a-tDCS. In addition, some research has reported positive effects
(i.e., flight time, muscular peak power and height) of the application of
a-tDCS with explosive movements such as the countermovement jump (CMJ) [38]. Even in protocols where high loads were
applied with incomplete recovery (e.g., 80% of the 1-RM with 60 s of recovery
between sets), the application of a-tDCS current allowed participants to
perform a larger number of repetitions compared to the placebo group. Likewise,
recent research [20] has shown significant
differences on EI values (*p* < 0.05) after the application of a-tDCS
compared to the Sham condition.

As in the present study, other studies have found no positive effects of a-tDCS on strength development in healthy participants [23], nor relevant effects on jumping performance [39], and also failed to show the effectiveness in improving sprint performance or reducing RPE during repeated sprint tasks [40]. Assuming that the ability to generate force is dependent on neural activity (i.e., synchronisation of motor units, as well as intermuscular coordination) and assuming that a-tDCS could facilitate this situation [41], further research is warranted to provide a better understanding of the effects of a-tDCS on diverse training programs and populations.

It should be noted that the effect of the a-tDCS application on sEMG activity has been less studied. Our analysis of muscle activity showed a higher initial activity in the RF of the right leg (area where the anodic stimulus was located, and dominant leg of all the participants) compared to the Sham group (*p* < 0.01). Similarly, Kamali et al. [42] have reported not only beneficial effects on sEMG signal after the application of a-tDCS (2 mA) for 13 min on the primary motor cortex (M1) and temporal cortex (TC) in bodybuilders, but also an improvement in performance (muscular strength and endurance). These improvements on neuromuscular performance have also been confirmed in previous research conducted in other types of muscle contractions, such as an isometric squat, where significant improvements in MVIC were shown [43]. Similar results have been reported in the biceps brachii muscles [41].

However, there are also contradictory results for neuromuscular performance changes, since no sEMG data were simultaneously collected. Several attempts have been made to demonstrate the effect in different populations, but no effect was found in a larger population [44]. In isokinetic lower-limb tasks with 2 and 4 mA stimuli, significant reductions in sEMG activity were observed [45]. Interestingly, current evidence suggests that the force cross-transfer effects following unilateral training may be modulated by increased corticospinal excitability of the ipsilateral primary motor cortex, possibly due to cross-activation [8]. Therefore, the effect of the application of a-tDCS on the untrained counterpart limb has been investigated, showing how a single session of a-tDCS combined with unilateral strength training of the right limb increases maximal strength and cross-activation in the untrained contralateral limb [22]. The current consensus is that the effect of a-tDCS lasts for 30 min post-application, with lower intra-individual than inter-individual variability [46]. This effect could be of interest when looking for post-activation potentiation performance enhancement, the improvement of which has been studied using a variety of techniques [47,48]. However, it has recently been shown that the use of a-tDCS for 15 min did not improve CMJ in young trained participants [40].

More recent studies [38,39] have shown contradictory results after the application of tDCS in regards to force production and sEMG activity. Whereas improvements on the rate of force development (RFD) and MVIC were reported in the non-dominant limb, no differences were found in the dominant leg [1]. Therefore, further research about this issue is warranted, not only on the specific dose of a-tDCS application, but also on different muscle contraction modalities, exercise selections, and athlete/patient profiles.

## 5. Limitations

Our study had some limitations that should be mentioned. First, this might be considered as an exploratory study due to the low sample size; however, the calculation of statistical power helped to draw accurate conclusions in a given population. Second, there are several differences in the a-tDCS protocols used to date, and even more so with respect to the Sham stimulation [31], which, in turn, makes comparisons with other studies difficult, and requires new studies to establish differences between a-tDCS protocols. Hence, we encourage readers to interpret and generalise our results with special caution, since further research is needed. Regarding sEMG results, the large standard deviations of the signals could be a consequence of the artefacts due to dynamic muscle contractions. Finally, we cannot extrapolate our results to patients immersed in a rehabilitation or physical reconditioning programme, given that healthy and strength-trained subjects participated in this study.

## 6. Conclusions

The application of a-tDCS over the DLPFC before exercise might not elicit positive effects on movement velocity or fatigue tolerance in strength-trained individuals. However, significant differences in right-leg muscle activity both in a-tDCS and Sham were reported. More research is needed to compare the effects of a-tDCS on strength-related variables in beginners, intermediate, or advanced participants. Since increasing the intensity of the sEMG signal is one of the priority objectives in any rehabilitation [49] or motor control improvement process [50], the results obtained in our research support the fact that a-tDCS may be considered as a complementary strategy in retraining and reconditioning physical therapy, but we are aware that further studies are warranted.

## Figures and Tables

**Figure 1 jcm-11-05220-f001:**
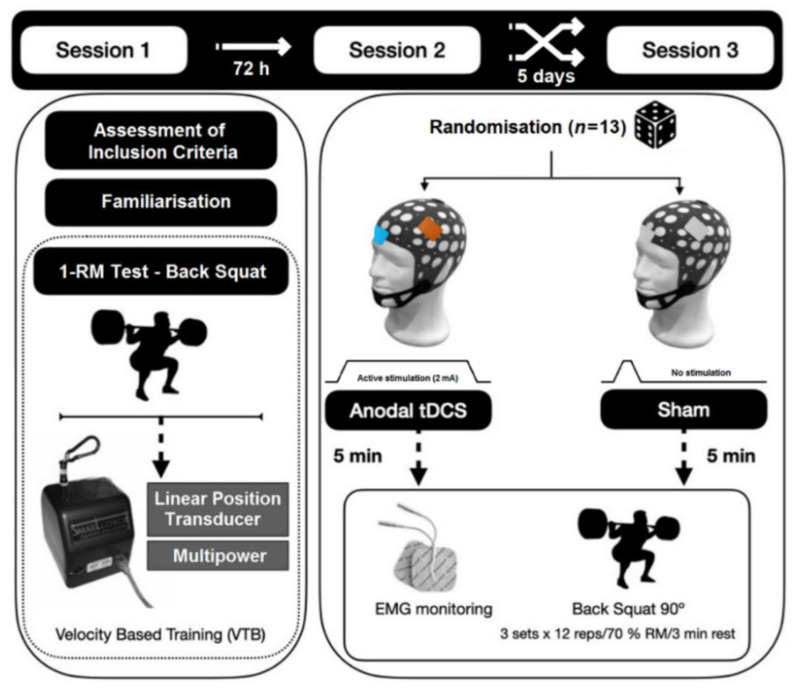
Scheme of the experimental design.

**Figure 2 jcm-11-05220-f002:**
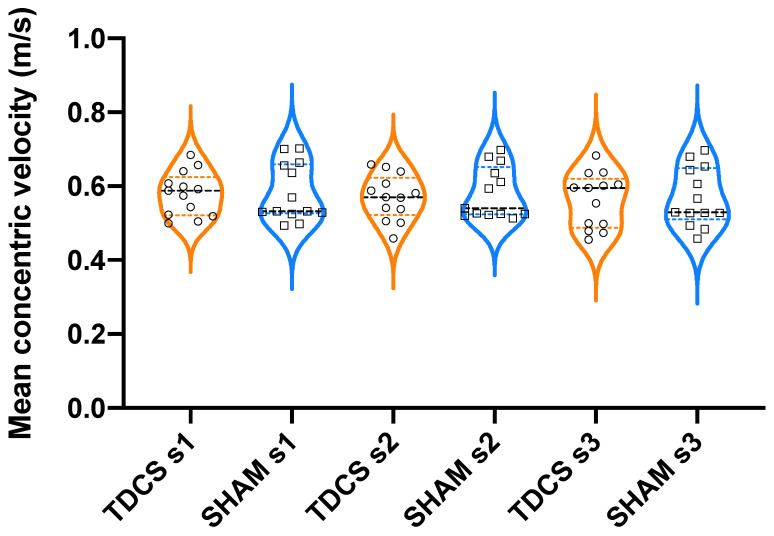
Violin-plots of mean velocity in anodal transcranial direct stimulation (a-TDCS) and Sham groups.

**Figure 3 jcm-11-05220-f003:**
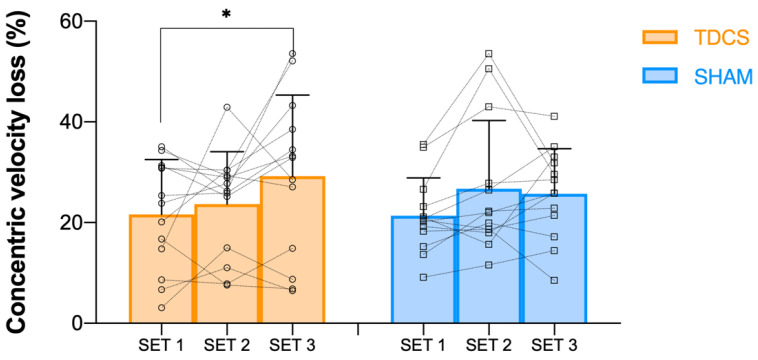
Concentric velocity loss (%) in set 1, set 2, and set 3 for both a-TDCS (orange) and Sham (blue) conditions, as well as individual responses for each set and condition. *: a significant (*p* < 0.05) difference from set 1 value.

**Figure 4 jcm-11-05220-f004:**
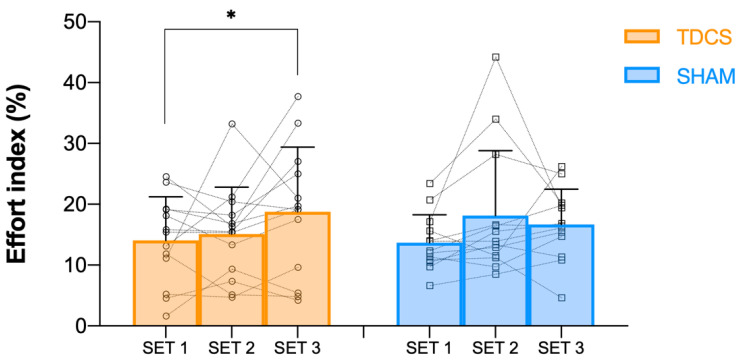
Effort index (%) in set 1, set 2, and set 3 for both a-TDCS (orange) and Sham (blue) conditions, as well as individual responses for each set and condition. *: a significant (*p* < 0.05) difference from set 1 value.

**Table 1 jcm-11-05220-t001:** The characteristics of the participants.

Variable	Mean ± SD
Age (years)	34.7 ± 3.3
Stature (cm)	178.0 ± 7.6
Body mass (kg)	76.8 ± 11.2
BMI (kg·m^−2^)	26.8 ± 4.2
1-RM (kg)	141.5 ± 16.3

Body Max Index (BMI), Standard Deviation (SD) and 1 Repetition Maximun (RM).

**Table 2 jcm-11-05220-t002:** Rectus femoris and vastus medialis EMGrms in the a-tDCS and Sham conditions.

	SET 1		SET 2		SET 3	
	Right Leg	Left Leg	Right Leg	Left Leg	Right Leg	Left Leg
**a-tDCS**						
Rectus Femoris (µV)	272.1 ± 86.6 *	283.1 ± 135.2	279.7 ± 80.3 *	282.0 ± 138.0	237.9 ± 104.1	228.0 ± 135.5
Vastus Lateralis (µV)	202.4 ± 69.2	213.6 ± 73.6	212.0 ± 60.9	208.1 ± 73.0	170.3 ± 80.7	153.8 ± 83.2
**Sham**						
Rectus Femoris (µV)	180.0 ± 47.6	267.1 ± 143.1	213.9 ± 72.1	285.0 ± 176.8	176.5 ± 63.3	246.6 ± 143.5
Vastus Lateralis (µV)	197.3 ± 67.8	218.2 ± 73.4	216.1 ± 79.4	232.9 ± 93.1	193.0 ± 84.2	203.6 ± 88.1

Data expressed as mean ± SD for both right and left legs during set 1, set 2, and set 3 of the BS exercise. * Significant difference between a-tDCS and Sham conditions (*p* < 0.05).

## Data Availability

The data that support the findings of this study are available upon request from the corresponding author.

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
