# Peer review of "Effects of Preceding Transcranial Direct Current Stimulation on Movement Velocity and EMG Signal during the Back Squat Exercise"

_jcm, 2022, doi:10.3390/jcm11175220_

Round 1

Reviewer 1 Report

GENERAL COMMENTS

In general, the topic of the article is very interesting and has a meaningful contribution to the field of research. But there are a few objectives and my personal opinions that should be taken into consideration by the author before considering it for publication.

The comments, broken down by manuscript sections, can be found below.

Generally,

the manuscript should be checked for grammar, spelling, and typos.

I suggest that “Preceding” Transcranial Direct Current Stimulation term is used to be clear that the stimulation was not applied during the squatting.

Title: Effects of Preceding Transcranial Direct Current Stimulation on Movement Velocity and EMG Signal During the Back Squat Exercise

ABSTRACT

19 – not during, it was applied before the squats, maybe: “Preceding Transcranial Direct Current Stimulation”

25 – how were the squats performed? All out in the concentric part of the squat? Or all-out in the eccentric and concentric part of the squat? Please explain also in the methods section.

INTRODUCTION

line 53 – reference style

line 55, 58 – merge citations

69 – typo, space

76 - the force-velocity relationship parameters, the a-tDCS allowed to maintain higher movement velocities while reducing the rate of perceived exertion (RPE). – add a citation

This seems to be the main application: endurance, strength endurance, not maximal strength. Please highlight/focus more on this information.

80 – typo, no space

80-81 - (i.e., force-velocity profile and sEMG signal) – force-velocity was not analysed, only velocity and velocity drop. This has no relationship with force-velocity profiling, more with velocity-based training. Therefore, this information is redundant. Please add variables that you have analysed. Moreover, explain background of the hypothesis. Why do you expect that? Mention 70% 1RM squats, multiple sets and sEMG of both legs.

Answer to this hypothesis based on the results in the first paragraph of the Discussion.

METHODS

87 – why firefighters?

90 – how was leg dominance determined?

90 – PARQ (explain abbreviation)

95 – were all firefighters male? If so, please add the information. If not, please add participants' characteristics, filtered by sex, into the table 1.

Table 1: Stature - maybe height? (optional)

125 – »less than 0.70 m∙s-1« add reference

132 – maybe “>speed 1.15 m∙s-1”

118 – Did “squat 1-RM” squat protocol consist of eccentric and concentric part of the squat, or only concentric one? Please add information.

134 – considered for analysis?

136 – Was reported by the participants. Please explain what questions were participants asked to get feedback on EI and RPE. When were these questions asked... at the end of the set, at the beginning of the next set? Please explain in detail, while one of the main outcomes of your study being EI and RPE.

118-136 – the protocol should be described more clearly.

For example:

               … at a gentle self-selected pace, five minutes of lower limb joint mobilization exercises, three 30 m running accelerations, followed by 3 sets of 5 repetitions of half squat with fixed weights of 20, 30 and 40 kg, the 1-RM test was performed in a multipower device (Element + Multipower, Technogym Trade Company 2017, Cesena, Italy) using a progressive load test (26)…

Please, use shorter sentences.

Was progressive load test a part of a warm-up procedure? Why did you use linear encoder at 1-RM test?

142 was begun – consider rewording

148 – what is the difference between “mean concentric velocity of each set” and “mean propulsive velocity of each set”?

148 - considering the 12 repetitions of each set: consider rewording, maybe average of 12 repetitions? Use consistently trough the text.

150 – how was the concentric velocity loss calculated? Best repetition/last repetition?

190 – Tests - signals

186 – why did you decide to find motor points of the muscles? 172: You followed SENIAM recommendations for electrode placement. I believe that motor points should be avoided when placing EMG electrodes to monitor muscle activity. From the text, I assume that you placed electrodes above the motor points. Please explain.

195 – How did you synchronise the repetitions of squats with the EMG signal? When did the acquisition of the EMG signal start and when did it stop? You used a linear encoder. I believe that the vertical position of the linear encoder could be used to define the repetitions and the times for the start and end of the EMG acquisition.

Moreover, it is not valid to report only raw EMG signals. I believe that signals should be normalized to the EMG MVC value or M-wave amplitude. Please explain. Please see:

Sousa, A. S. P. & Tavares, J. M. R. S. (2012). Surface electromyographic amplitude normalization methods: A review. (H. Takada, Ur.), Electromyography: New Developments, Procedures, and Applications. Hauppaugeu: Nova Science Publishers, Inc. https://repositorio-aberto.up.pt/bitstream/10216/64430/2/67854.pdf

Moreover, you report very large standard deviations of the signals. For example, 176.5 ± 63.3, 246.6 ± 143.5 (right, left leg, SET 3, SHAM, RF). I believe this is a consequence of the artefacts due to dynamic muscle contractions. Please mention this in the limitations section. Due to a large standard deviation, it is difficult to expect any differences between the sets and legs.

194 – The stimulation did not occur simultaneously with the exercise. It occurred before the exercise had begun. Please correct accordingly in the text.

202-204 – some information is missing. What?

Nevertheless, I think you should have used mixed-model ANOVA (one within and one between factor).

208 – cite other studies

215 – main effects

221 – set interaction? main effect of set

Figure 2 – add notes/abbreviations. TDCS s1 – should be explained in detail

Figure 3 – abbreviations should be below the figure

RESULTS

Results are confusing. When using two-way ANOVA (again, mixed-model I think) you should report interactions and then main effects of intervention and set. Then, post-hoc pairwise comparisons should be reported in the case that interaction or one of the main effects were statistically significant.

For example:

“ … In addition, a significant main effect of set on right leg VM muscle activity (p<0.05, F=6.49) was observed between set 1 and set 2, which indicated a higher VM muscle activity in the Sham condition when set 2 was compared to set 1 (p<0.05; F = 4.48, 9.5%). No significant differences between legs were observed for any other variable… “

Did you mean the main effect of the set, or perhaps the set*leg interaction? (Here you used a 3-way anova). If you meant the main effect of the set, a post-hoc analysis would show the differences between the sets. The leg and intervention factors have only two levels, so no additional post-hoc analysis is needed.

“No significant differences between legs were observed for any other variable… “ – did you get a main effect of the factor leg? If not, why did you do pairwise comparisons at each set?

The results should be written more clearly.

DISCUSSION

251 – the force-velocity relationship was not analyzed. I think this information is redundant. Please correct through the text. You used a linear encoder for other purposes, not plotting F-v profiles.

253 – what was the tempo of the squat execution? As fast as possible at each repetition in the set? This should be clearly defined in the methods.

258 – based on what can you state that? The mechanism should be explained. This is a strong claim.

261 – capital letter

283 – you did not measure maximal strength

293 – this should be mentioned in the introduction. Why do you expect difference between the legs in the EMG activity after the stimulation?

300 – were shown

Can EMG amplitude be used to monitor fatigue (central, peripheral)? Was the response to fatigue higher in the a-tDCS group? Can this be related to the a-tDCS protocol 

Author Response

REVIEWER 1

GENERAL COMMENTS

In general, the topic of the article is very interesting and has a meaningful contribution to the field of research. But there are a few objectives and my personal opinions that should be taken into consideration by the author before considering it for publication.

The comments, broken down by manuscript sections, can be found below.

Generally,

the manuscript should be checked for grammar, spelling, and typos.

Authors’ response: We would like to thank the editor for taking time to provide last comments to improve the quality of the manuscript. Taking into account your recommendations, the points you have made have been modified in the corresponding parts of the text, in the hope that the changes will be to your liking.

I suggest that “Preceding” Transcranial Direct Current Stimulation term is used to be clear that the stimulation was not applied during the squatting.

Title: Effects of Preceding Transcranial Direct Current Stimulation on Movement Velocity and EMG Signal During the Back Squat Exercise

Author's response: Thank you for your comments. It has been modified accordingly.

ABSTRACT

19 – not during, it was applied before the squats, maybe: “Preceding Transcranial Direct Current Stimulation”

Author's response: Thank you for your comments. It has been modified including the recommended term.

25 – how were the squats performed? All out in the concentric part of the squat? Or all-out in the eccentric and concentric part of the squat? Please explain also in the methods section.

 Author's response: Thank you for your comments. It has been modified including (lines 147-149).

INTRODUCTION

line 53 – reference style

Author's response: Thank you for your comments. It has been modified accordingly.

line 55, 58 – merge citations

Author's response: Thank you for your comments. It has been modified accordingly.

69 – typo, space

Author's response: Thank you for your comments. It has been modified accordingly.

76 - the force-velocity relationship parameters, the a-tDCS allowed to maintain higher movement velocities while reducing the rate of perceived exertion (RPE). – add a citation

Author's response: Thank you for your comments. It has been modified accordingly, including a citation.

This seems to be the main application: endurance, strength endurance, not maximal strength. Please highlight/focus more on this information.

80 – typo, no space

Author's response: Thank you for your comments. It has been modified accordingly.

80-81 - (i.e., force-velocity profile and sEMG signal) – force-velocity was not analysed, only velocity and velocity drop. This has no relationship with force-velocity profiling, more with velocity-based training. Therefore, this information is redundant. Please add variables that you have analysed. Moreover, explain background of the hypothesis. Why do you expect that? Mention 70% 1RM squats, multiple sets and sEMG of both legs.

Answer to this hypothesis based on the results in the first paragraph of the Discussion.

Author's response: Thank you for your comments. It has been modified accordingly including all the above-mentioned concepts.

METHODS

87 – why firefighters?

Authors’ response: Firefighters fit the profile of highly trained subjects with good exercise technique to be analysed.

90 – how was leg dominance determined?

Authors’ response: Self – reported appearance (included in the text).

90 – PARQ (explain abbreviation)

Author's response: Thank you for your comments. It has been included accordingly.

95 – were all firefighters male? If so, please add the information. If not, please add participants' characteristics, filtered by sex, into the table 1.

Author's response: Thank you for your comments. Information has been included in the participants section to clarify this concept (line 90).

Table 1: Stature - maybe height? (optional)

125 – »less than 0.70 m∙s-1« add reference

Author's response: Thank you for your comments. It has been modified accordingly, including a citation.

132 – maybe “>speed 1.15 m∙s-1”

Author's response: Thank you for your comments. It has been modified accordingly.

118 – Did “squat 1-RM” squat protocol consist of eccentric and concentric part of the squat, or only concentric one? Please add information.

Author's response: Thank you for your comments. Text has been included clarifying the focus on the concentric phase (line 144).

134 – considered for analysis?

Author's response: Thank you for your comments. The effort index is analysed as one of the variables (Figure 4).

136 – Was reported by the participants. Please explain what questions were participants asked to get feedback on EI and RPE. When were these questions asked... at the end of the set, at the beginning of the next set? Please explain in detail, while one of the main outcomes of your study being EI and RPE.

Author's response: Thank you for your comments. The effort index was calculated by multiplying the concentric velocity loss of the set by the mean concentric velocity of the best repetition. RPE was asked of each participant at the end of each series. As no difference was found, and in order not to confuse the reader, we proceed to eliminate this part of the text.

118-136 – the protocol should be described more clearly.

For example:

               … at a gentle self-selected pace, five minutes of lower limb joint mobilization exercises, three 30 m running accelerations, followed by 3 sets of 5 repetitions of half squat with fixed weights of 20, 30, and 40 kg, the 1-RM test was performed in a multipower device (Element + Multipower, Technogym Trade Company 2017, Cesena, Italy) using a progressive load test (26)…

Please, use shorter sentences.

Author's response: Thank you for your comments. We have modified the grammar to improve the comprehension of the section.

Was progressive load test a part of a warm-up procedure? Why did you use linear encoder at 1-RM test?

Author's response: Thank you for your comments. No, those three loads were the final part of the warm-up (this has been clarified in the text). As far as the use of the encoder is concerned, it is a very well thought-out, fast, reliable way, and moreover, it is the same device that was later used in the tDCS and Spam sessions.

142  was begun – consider rewording

Author's response: Thank you for your comments. It has been modified accordingly.

148 – what is the difference between “mean concentric velocity of each set” and “mean propulsive velocity of each set”?

Author's response: Thank you for your comments. Mean concentric velocity data include the braking phase, mean propulsive velocity shows the acceleration phase.

148 - considering the 12 repetitions of each set: consider rewording, maybe average of 12 repetitions? Use consistently trough the text.

Author's response: Thank you for your comments. It has been modified accordingly.

150 – how was the concentric velocity loss calculated? Best repetition/last repetition?

Author's response: Thank you for your comments. It was indeed calculated in this way. The best repetition was usually the first in all subjects. We have included this comment in the text.

190 – Tests – signals

Author's response: Thank you for your comments. It has been modified accordingly.

186 – why did you decide to find motor points of the muscles? 172: You followed SENIAM recommendations for electrode placement. I believe that motor points should be avoided when placing EMG electrodes to monitor muscle activity. From the text, I assume that you placed electrodes above the motor points. Please explain.

Author's response: Thank you for your comments: SENIAM recommendations were followed. Only the motor points were used as a safety guide. This part of the text has been removed to avoid confusion.

195 – How did you synchronise the repetitions of squats with the EMG signal? When did the acquisition of the EMG signal start and when did it stop? You used a linear encoder. I believe that the vertical position of the linear encoder could be used to define the repetitions and the times for the start and end of the EMG acquisition.

Author's response: Thank you for your comments. It has been clarified accordingly.

Moreover, it is not valid to report only raw EMG signals. I believe that signals should be normalized to the EMG MVC value or M-wave amplitude. Please explain. Please see:

Sousa, A. S. P. & Tavares, J. M. R. S. (2012). Surface electromyographic amplitude normalization methods: A review. (H. Takada, Ur.), Electromyography: New Developments, Procedures, and Applications. Hauppaugeu: Nova Science Publishers, Inc. https://repositorio-aberto.up.pt/bitstream/10216/64430/2/67854.pdf

Moreover, you report very large standard deviations of the signals. For example, 176.5 ± 63.3, 246.6 ± 143.5 (right, left leg, SET 3, SHAM, RF). I believe this is a consequence of the artefacts due to dynamic muscle contractions. Please mention this in the limitations section. Due to a large standard deviation, it is difficult to expect any differences between the sets and legs.

Author's response: Thank you for your comments. It has been modified accordingly. Thus, the amplitude of the temporally processed electromyography can only be used to assess short-term changes in the activity of a single muscle from the same individual when the electrode setup has not been altered.

194 – The stimulation did not occur simultaneously with the exercise. It occurred before the exercise had begun. Please correct accordingly in the text.

Authors’ response: Thank you for your comments. It has been clarified accordingly.

202-204 – some information is missing. What?

Nevertheless, I think you should have used mixed-model ANOVA (one within and one between factor).

Author's response: Thank you for your comments. We very much appreciate the reviewer’s comments and suggestions regarding results section. On the light’s of reviewer’s suggestion, we have modified the results section accordingly. Please see pages 6-8.

208 – cite other studies

215 – main effects

221 – set interaction? main effect of set

Figure 2 – add notes/abbreviations. TDCS s1 – should be explained in detail

Figure 3 – abbreviations should be below the figure

RESULTS

Results are confusing. When using two-way ANOVA (again, mixed-model I think) you should report interactions and then main effects of intervention and set. Then, post-hoc pairwise comparisons should be reported in the case that interaction or one of the main effects were statistically significant.

For example:

“ … In addition, a significant main effect of set on right leg VM muscle activity (p<0.05, F=6.49) was observed between set 1 and set 2, which indicated a higher VM muscle activity in the Sham condition when set 2 was compared to set 1 (p<0.05; F = 4.48, 9.5%). No significant differences between legs were observed for any other variable… “

Did you mean the main effect of the set, or perhaps the set*leg interaction? (Here you used a 3-way anova). If you meant the main effect of the set, a post-hoc analysis would show the differences between the sets. The leg and intervention factors have only two levels, so no additional post-hoc analysis is needed.

“No significant differences between legs were observed for any other variable… “ – did you get a main effect of the factor leg? If not, why did you do pairwise comparisons at each set?

The results should be written more clearly.

DISCUSSION

251 – the force-velocity relationship was not analyzed. I think this information is redundant. Please correct through the text. You used a linear encoder for other purposes, not plotting F-v profiles.

Author's response: Thank you for your comments. It has been clarified accordingly by mentioning movement velocity instead of force-velocity profile

253 – what was the tempo of the squat execution? As fast as possible at each repetition in the set? This should be clearly defined in the methods.

Author's response: Thank you for your comments. It has been clarified accordingly.

258 – based on what can you state that? The mechanism should be explained. This is a strong claim.

Author's response: Thank you for your comments. Our results show this evolution of the variables IE and velocity loss (figure 3 and 4). The mechanism that may explain this is the depolarising neural soma and hyperpolarised apical dendrite (Hamada et al.2012).

261 – capital letter

Author's response: Thank you for your comments. It has been clarified accordingly.

283 – you did not measure maximal strength

Authors’ response: Thank you for your comments. It has been clarified accordingly.

293 – this should be mentioned in the introduction. Why do you expect the difference between the legs in the EMG activity after the stimulation?

Author's response: Thank you for your comments. Previous research has shown changes in this variable in lower limb strength tasks. (Kamali A-M, Saadi ZK, Yahyavi S-S, Zarifkar A, Aligholi H, NamiM (2019) Transcranial direct current stimulation to enhance athletic performance outcome in experienced bodybuilders. PLoS ONE 14(8): e0220363. https:// doi.org/10.1371/journal.pone.0220363) This concept is expanded upon in the introduction to the text

300 – were shown

Authors’ response: Thank you for your comments. It has been modified accordingly.

Can EMG amplitude be used to monitor fatigue (central, peripheral)? Was the response to fatigue higher in the a-tDCS group? Can this be related to the a-tDCS protocol 

Authors’ response: Thank you for your comments. Surface EMG can indeed be a method for fatigue estimation, central o peripheral (Córdova A, Nuin I, Fernández-Lázaro D, Latasa I, Rodríguez-Falces J. Actividad electromiográfica (EMG) durante el pedaleo, su utilidad en el diagnóstico de la fatiga en cidistas. Arch Med del Deport. 2017;34(4):217–23). Probably, the changes between the Sham session and a-tDCS were due to the effects of the application of the technique.

Reviewer 2 Report

Thank you very much for the opportunity to review the manuscript submission entitled: "Effects of Transcranial direct current stimulation on movement velocity and EMG signal during the back squat exercise." The current manuscript aims to evaluate the effects of anodal Transcranial Direct Stimulation (tDCS) over the dorsolateral prefrontal cortex (DLPFC) during the back squat exercise on movement velocity and surface electromyographic (sEMG) activity in thirteen healthy well-trained male firefighters. The manuscript is well structured and the methodology is described in detail. However, some considerations should be taken into account to improve the quality of the study. The introduction and discussion section should be improved.

Abstract

The abstract should be a total of about 200 words maximum according to the instructions to authors provided by Journal of Clinical Medicine.

Introduction

Please include a reference to support this statement (P1 L40-41).

Most of the tDCS studies focus on M1 or DLPFC. It would be interesting if the authors would expand on this issue in the introduction section for a better understanding of the choice of the application region.

Please define abbreviation DC (P2 L48).

Please check the reference format (P2 L53).

Please define abbreviation RT (P2 L59).

There are some abbreviations that could be eliminated as they are not mentioned again in the manuscript (TTE, MC) or if the authors decide to use the acronym indicated in the introduction, they should use it throughout the manuscript.

Highlight the contribution of your study to the literature.

Authors might consider stating the objective of the study and the hypotheses in a final paragraph of the introduction for better comprehension of the purpose of the study.

Materials and methods

Participants

The PARQ and IPAQ instruments should be included and defined in the Materials and methods section and the most relevant information obtained could be indicated in Table 1.

The sample size could be indicated in this section and eliminated from the statistical analysis for better comprehension.

Please indicate the location/facilities where the measurements took place.

Please check the reference format (P3 L104).

Given that the job of firefighter involves long working schedules. Were the effects on participants' sleep monitored? Could this aspect affect strength production in any way?

No data were obtained on the rate of perceived exertion that could be added to the analyses of the study?

Back Squat Exercise Protocol

The protocol described by the authors in the manuscript is 3 sets of 12 repetitions with a 2-minute rest period between sets (P4 L141). However, the Figure 1 shows 3 sets of 12 repetitions with a 3-minute rest period between sets, please check this issue.

Looking at Figure 1 it is understood that after the application of a-tDCS or SHAM there was 5 minutes of rest before starting the back squat exercise protocol. If this is the case the authors should indicate this issue in the manuscript for better understanding.

a-tDCS procedures

Please, specify the position the participant adopted during stimulation

Statistical analysis

Authors should use the format: mean (SD), not mean ± SD according to the General Principles for Reporting Statistical Results specified in the The SAMPL Guidelines.

Results

Authors should use the format: mean (SD), not mean ± SD in Table 2. (Take this information into account for the abstract).

Discussion

It would be interesting to indicate in the first paragraph whether or not the hypotheses set in the study were fulfilled and from this point to establish the discussion with respect to the literature.

Please check although (P8 L261).

When the authors refer to the ergogenic effects obtained from a-TDCS, the references that support this claim should be indicated.

Reference 38 is a systematic review and meta-analysis. Please verify that the specified information is correct.

The authors should provide further discussion when comparing the results of the study with the literature by indicating the area of stimulus application, intensity, timing of application and recovery, duration of the stimulus.

The penultimate and the last one paragraph of the discussion should be revised. The information should be restructured for a better understanding that will reach the reader easily. In addition, references 51 and 53 do not seem to report information on post-activation potentiation performance enhancement (the citation is not complete and it is difficult to consult them).

The authors may wish to conclude the discussion with a paragraph of future directions.

Clearly highlight the implications of the results of the study.

Conclusions

Conclusions should be established based on the aims of the study. There is information in this section that could be included in the discussion section as future directions.

References

Please, check that the references are complete and well listed.

General considerations

Please check if abbreviations appearing in the text are necessary in some cases.

Author Response

Comments and Suggestions for Authors

Thank you very much for the opportunity to review the manuscript submission entitled: "Effects of Transcranial direct current stimulation on movement velocity and EMG signal during the back squat exercise." The current manuscript aims to evaluate the effects of anodal Transcranial Direct Stimulation (tDCS) over the dorsolateral prefrontal cortex (DLPFC) during the back squat exercise on movement velocity and surface electromyographic (sEMG) activity in thirteen healthy well-trained male firefighters. The manuscript is well structured and the methodology is described in detail. However, some considerations should be taken into account to improve the quality of the study. The introduction and discussion section should be improved.

 Authors’ response: Thank you for your comments. Changes have been made and the text has been adapted based on your suggestions as follows.

Abstract

The abstract should be a total of about 200 words maximum according to the instructions to authors provided by Journal of Clinical Medicine.

Authors’ response: Thank you for your comments. It has been modified accordingly, by conforming to the standards.

Introduction

Please include a reference to support this statement (P1 L40-41).

Authors’ response: Thank you for your comments. It has been modified accordingly (Lu P, Hanson NJ, Wen L, Guo F and Tian X (2021) Transcranial Direct Current Stimulation Enhances Muscle Strength of Non-dominant Knee in Healthy Young Males. Front. Physiol. 12:788719. doi: 10.3389/fphys.2021.788719

Most of the tDCS studies focus on M1 or DLPFC. It would be interesting if the authors would expand on this issue in the introduction section for a better understanding of the choice of the application region.

Authors’ response: Thank you for your comments. The required concept has been extended.

Please define abbreviation DC (P2 L48).

Authors’ response: Thank you for your comments. It has been modified accordingly.

Please check the reference format (P2 L53).

Authors’ response: Thank you for your comments. It has been modified accordingly.

Please define abbreviation RT (P2 L59).

Authors’ response: Thank you for your comments. It has been modified accordingly.

There are some abbreviations that could be eliminated as they are not mentioned again in the manuscript (TTE, MC) or if the authors decide to use the acronym indicated in the introduction, they should use it throughout the manuscript.

Authors’ response: Thank you for your comments. It has been modified accordingly.

Highlight the contribution of your study to the literature.

Authors’ response: Thank you for your comments. this concept has been expanded in the introductory section.

Authors might consider stating the objective of the study and the hypotheses in a final paragraph of the introduction for better comprehension of the purpose of the study.

Authors’ response: Thank you for your comments. It has been modified accordingly.

Materials and methods

Participants

The PARQ and IPAQ instruments should be included and defined in the Materials and methods section and the most relevant information obtained could be indicated in Table 1.

Authors’ response: Thank you for your comments. It has been modified accordingly expanding on the information relating to the results obtained in these questionnaires.

The sample size could be indicated in this section and eliminated from the statistical analysis for better comprehension.

Authors’ response: Thank you for your comments. We very appreciate reviewer’s comments and suggestions regarding results section. On the light’s of reviewer’s suggestion, we have modify the results section accordingly. Please see pages 6-8.

Please indicate the location/facilities where the measurements took place.

Authors’ response: Thank you for your comments. It has been modified accordingly.

Please check the reference format (P3 L104).

Authors’ response: Thank you for your comments. It has been modified accordingly.

Given that the job of firefighter involves long working schedules. Were the effects on participants' sleep monitored? Could this aspect affect strength production in any way?

Authors’ response: Thank you for your comments. Sleep quality was not monitored. It is true, however, that none of the participants had any emergency actions during the duration of the measurements, which could guarantee a normal situation in terms of fatigue, or at least avoid excessive fatigue of the participants. This could certainly have been a variable to be taken into account.

No data were obtained on the rate of perceived exertion that could be added to the analyses of the study?

 Authors’ response: Thank you for your comments. We found no difference between the two conditions (Sham/a-tDCS) among the participants.

Back Squat Exercise Protocol

The protocol described by the authors in the manuscript is 3 sets of 12 repetitions with a 2-minute rest period between sets (P4 L141). However, the Figure 1 shows 3 sets of 12 repetitions with a 3-minute rest period between sets, please check this issue.

Authors’ response: Thank you for your comments. It has been modified accordingly.

Looking at Figure 1 it is understood that after the application of a-tDCS or SHAM there was 5 minutes of rest before starting the back squat exercise protocol. If this is the case the authors should indicate this issue in the manuscript for better understanding.

 Authors’ response: Thank you for your comments. It has been modified accordingly expanding on the information relating to this item.

a-tDCS procedures

Please, specify the position the participant adopted during stimulation

 Authors’ response: Thank you for your comments. It has been modified accordingly expanding on the information relating to this item.

Statistical analysis

Authors should use the format: mean (SD), not mean ± SD according to the General Principles for Reporting Statistical Results specified in the The SAMPL Guidelines.

Authors’ response: Thank you for your comments. We very appreciate reviewer’s comments and suggestions regarding results section. On the light’s of reviewer’s suggestion, we have modify the results section accordingly. Please see pages 6-8.

Results

Authors should use the format: mean (SD), not mean ± SD in Table 2. (Take this information into account for the abstract).

Authors’ response: Thank you for your comments. We very appreciate reviewer’s comments and suggestions regarding results section. On the light’s of reviewer’s suggestion, we have modify the results section accordingly. Please see pages 6-8.

Discussion

It would be interesting to indicate in the first paragraph whether or not the hypotheses set in the study were fulfilled and from this point to establish the discussion with respect to the literature.

 Authors’ response: Thank you for your comments. It has been modified accordingly expanding on the information relating to this item.

Please check although (P8 L261).

Authors’ response: Thank you for your comments. It has been modified accordingly.

When the authors refer to the ergogenic effects obtained from a-TDCS, the references that support this claim should be indicated.

Authors’ response: Thank you for your comments. The citation has been included.

Reference 38 is a systematic review and meta-analysis. Please verify that the specified information is correct.

Authors’ response: Thank you for your comments. This is the correct citation (Park S-B, Sung DJ, Kim B, Kim S, Han J- K (2019) Transcranial Direct Current Stimulation of motor cortex enhances running performance. PLoS ONE 14(2): e0211902. https://doi.org/ 10.1371/journal.pone.0211902)

The authors should provide further discussion when comparing the results of the study with the literature by indicating the area of stimulus application, intensity, timing of application and recovery, duration of the stimulus.

 Authors’ response: Thank you for your comments. It has been modified accordingly expanding on the information relating to this item.

The penultimate and the last one paragraph of the discussion should be revised. The information should be restructured for a better understanding that will reach the reader easily. In addition, references 51 and 53 do not seem to report information on post-activation potentiation performance enhancement (the citation is not complete and it is difficult to consult them).

Authors’ response: Thank you for your comments. This section has been rewritten for the sake of clarity in the presentation of the arguments.

The authors may wish to conclude the discussion with a paragraph of future directions.

Authors’ response: Thank you for your comments. It has been modified accordingly.

Clearly highlight the implications of the results of the study.

 Authors’ response: Thank you for your comments. It has been modified accordingly.

Conclusions

Conclusions should be established based on the aims of the study. There is information in this section that could be included in the discussion section as future directions.

Authors’ response: Thank you for your comments. It has been modified accordingly.

References

Please, check that the references are complete and well listed.

Authors’ response: Thank you for your comments. It has been modified accordingly.

General considerations

Please check if abbreviations appearing in the text are necessary in some cases.

Authors’ response: Thank you for your comments. It has been modified accordingly.

Round 2

Reviewer 2 Report

Abstract

Again, the authors exceeded the 200-word limit in the abstract as indicated in the instructions to authors provided by Journal of Clinical Medicine. Please correct this issue.

Introduction

I think a more accurate and specific reference for the first sentence of your manuscript could be a meta-analysis, systematic review or similar. This would further reinforce the current state of knowledge.

I suggest including: Chinzara, T. T., Buckingham, G., & Harris, D. J. (2022). Transcranial direct current stimulation and sporting performance: A systematic review and meta‐analysis of transcranial direct current stimulation effects on physical endurance, muscular strength and visuomotor skills. European Journal of Neuroscience55(2), 468-486.

Specify in the aims of the study the stimulation region.

Materials and methods

Back squat one-repetition maximum test (BS 1-RM)

The 1-RM test was performed in a multipower device (Element + Multipower, Technogym Trade Company 2017, Cesena, Italy). Does this device include the linear encoder? If not, the authors should provide information on the device and, if available, a reliability study to support its use.

Surface EMG protocol

The authors have added the following valuable information in the text: “The recording of sEMG data was synchronised with the repetitions and the times for the start and end position of with the position transducer, recording each and every repetition performed by each participant”. It would be interesting to include the data synchronization unit or the procedure in which the authors carried out this synchronization so that it could be replicable in other studies.

Statistical analysis

Again, the authors should modify the section on abstract, statistical analysis, and results to use the format: mean (SD), not mean ± SD, according to the General Principles for Reporting Statistical Results specified in The SAMPL Guidelines.

Results

Although the authors have not obtained differences in RPE between the two types of stimulation, this information should be included in the manuscript since most of the research focusing on this topic has reported the RPE values obtained, which may be helpful for the development of future systematic reviews or meta-analyses.

The authors should modify the title of Table 2 to correspond to the muscles analyzed.

Discussion

Were the hypotheses fulfilled? Indicate clearly in the first paragraph of the discussion.

P9 L269 Vastus Lateralis

“Even though the application time (15-20 minutes), the intensity (≈2 mA) and the pre-dominant application areas (M1 and DLPFC) are very standardised, there is a great het-erogeneity in the applied exercise methodology”. Please, add references.

Conclusions

Again, the authors should establish the conclusions based on the aims of the study. There is information in this section that could be included in the discussion section as future directions.